# Death within 1 year among emergency medical admissions to Scottish hospitals: incident cohort study

Emily Moore,[1] Rosalia Munoz-Arroyo,[1] Lauren Schofield,[1] Alice Radley,[2] David Clark,[3] Chris Isles[2]

[1]NHS National Services Scotland, Edinburgh, UK
[2]Department of Medicine, Dumfries and Galloway Royal Infirmary, Dumfries, UK
[3]School of Interdisciplinary Studies, University of Glasgow, Glasgow, UK

**Correspondence to**
Dr Chris Isles;
chris.isles@nhs.net

## ABSTRACT

**Background** It is increasingly recognised that large numbers of hospital inpatients have entered the last year of their lives.

**Aim** To establish the likelihood of death within 12 months of admission to hospital; to examine the influence on survival of a cancer diagnosis made within the previous 5 years; to assess whether previous emergency admissions influenced mortality; and to compare mortality with that of the wider Scottish population.

**Design** Incident cohort study.

**Setting** 22 hospitals in Scotland.

**Participants** This study used routinely collected data from 10 477 inpatients admitted as an emergency to medicine in 22 Scottish hospitals between 18 and 31 March 2015. These data were linked to national death records and the Scottish Cancer Registry.

**Primary outcome measures** 1 year cohort mortality compared with that of the general Scottish population. Patient factors correlating with higher risk of mortality were identified using Cox regression.

**Results** There were 2346 (22.4%) deaths in the year following the census admission. Six hundred and ten patients died during that admission (5.8% of all admissions and 26% of all deaths) while 1736 died after the census admission (74% of all deaths). Malignant neoplasms (33.8%), circulatory diseases (22.5%) and respiratory disease (17.9%) accounted for almost three-quarters of all deaths. Mortality rose steeply with age and was five times higher at 1 year for patients aged 85 years and over compared with those who were under 60 years of age (41.9%vs7.9%) (p<0.001). Patients with cancer had a higher mortality rate than patients without a cancer diagnosis (55.6%vs16.6%) (p<0.001). Mortality was higher among patients with one or more emergency medical admissions in the previous year (30.1% v 15.1%) (p<0.001). Age/sex-standardised mortality was 110.4 (95% CI 104.4 to 116.5) for the cohort and 11.7 (95% CI 11.6 to 11.8) for the Scottish population, a 9.4-fold increase in risk.

**Conclusion** These data may help identify groups of patients admitted to hospital as medical emergencies who are at greatest risk of dying not only during admission but also in the following 12 months.

## Strengths and limitations of this study

► This was an incident rather than a prevalent cohort study.
► The source of all data related to hospital stays was the Scottish Morbidity Record and the Scottish Cancer Registry database.
► Linkage to National Records of Scotland death records allowed us to follow patients for a year and match time of death to the admission record.
► If patients emigrated and died abroad during the year of follow-up then this would underestimate their mortality. We think this is unlikely to be a source of major imprecision.
► We recognise that our analysis identifies groups of individuals at high risk of death within 1 year and that it cannot and should not be used to predict an individual's risk of dying.

indicating that, for many individuals, admission to hospital is a sentinel event marking the transition to the last year of their lives.[1 2] Nearly 1 in 10 patients died during admission, and almost 1 in 3 patients had died by a year later. This increased to nearly 1 in 2 for the over 85 age group. This information highlighted a need for clinicians to alter their approach to patient care in order to identify and address key end-of-life care needs.[1 2] Colleagues in Ireland[3] and New Zealand[4] have conducted similar analyses and drawn similar conclusions. This realistic, patient-focused approach has been widely advocated by the Gold Standards Framework,[5] National Institute for Health and Clinical Excellence,[6] the General Medical Council,[7] NHS England[8] and Health Improvement Scotland.[9]

Despite these clear recommendations, many clinicians are reluctant to address end-of-life issues. A National Confidential Enquiry into Patient Outcome and Death review of the care of patients who died within 30 days of receiving systemic anticancer therapy found that the decision to treat with chemotherapy was inappropriate in 19% cases. This raised

## INTRODUCTION

We have previously shown high 12-month mortality among Scottish hospital inpatients

questions as to whether patients with cancer are given enough information about chemotherapy to enable them to make an informed consent to treatment.[10] In a study of patients with metastatic lung and colorectal cancer, 69% and 81% of patients, respectively, were unaware that chemotherapy was highly unlikely to cure their cancer, again suggesting that clinicians are not comfortable with end-of-life care discussions.[11] A survey of over 4000 US physicians found that one-third would not discuss prognosis with a patient with cancer who was asymptomatic but had only 4–6 months to live, preferring instead to wait until symptoms developed or there were no more treatments to offer.[12]

Our previous study was of a prevalent rather than incident cohort which may have over-represented patients who had longer hospital stays. Likelihood of death was two times higher in medical patients than surgical patients, possibly reflecting the elective nature of most surgical admissions. We did not examine the influence of diagnosis, particularly a cancer diagnosis, as a predictor of death nor did we evaluate the relation between previous hospital admissions and mortality or the mortality risk of hospital inpatients compared with the wider population from which our patients were derived.[1 2] The aims of the current study, therefore, were to examine an incident rather than a prevalent cohort, to focus on patients admitted as emergencies to medicine, to determine the impact on survival of a cancer diagnosis made within the previous 5 years, to assess whether previous admissions to hospital influenced mortality and to compare mortality with that of an age/sex-standardised Scottish general population.

## METHODS

We included only patients admitted to hospitals in Scotland where the most acute clinical activity occurs: large general hospitals (n=15) and teaching hospitals (n=7). On this occasion, we limited our analyses to inpatients admitted as an emergency to medicine between 18 and 31 March 2015. We defined an inpatient (rather than a day case) as a person who had a Scottish Morbidity Record (SMR01) episode with a discharge date either the day following the admission date or later. Preliminary checks of numbers of patients and deaths in relation to demographic variables indicated that a 14-day census period resulted in selecting sufficiently large numbers of admissions and deaths for robust statistical analysis.

In the event of an emergency readmission within the 2-week census period (n=216), we only counted the patient once. Thus, our analysis is based on their first (or only) admission during this period. We refer to this as the census admission and the date of this admission as the census date. We classified patients as having a cancer diagnosis (any malignant neoplasm International Classification of Diseases, Tenth Edition code C00-C97, excluding non-melanoma skin cancer (C44))[13] if they had an SMR01 record with a cancer diagnosis or a record in the Scottish

Cancer Registry database[14] dated 5 years or less prior to their census date. We also included any cancer diagnoses made in the census stay.

The source of all data related to hospital stays was the SMR01.[15] The measure of deprivation used was the Scottish Index of Multiple Deprivation (SIMD).[16] This is an area-based deprivation score, which ranks areas according to a relative measure of deprivation where SIMD 1 represents 20% of the most deprived areas in Scotland and SIMD 5 represents 20% of the least deprived areas. Linkage to National Records of Scotland (NRS) death records[17] allowed us to follow patients for a year and match time of death to the admission record.[18] We limited the analysis to Scottish residents (n=42 persons with invalid or non-Scottish postcodes excluded) and excluded records where record linkage was not possible due to omissions or errors (n=46). We calculated mortality rates for the general Scottish population in 2015 from NRS death records and the NRS midyear population estimates for 2015.

### Statistical analysis

We provided statistical summaries in relation to potential risk factors for deaths in the follow-up year, including mortality at 7 days, 30 days, 3 months, 6 months, 9 months and 12 months from the census date. Risk factors were patient demographics (age, sex and deprivation) having a cancer diagnosis (see Methods for definition) and whether admitted as an emergency in the year prior to the census admission. We classified primary diagnoses at census admission and primary causes of deaths using the NRS classification for causes of death in Scotland[19] and documented whether death occurred in a hospital, a care home or other institution or at a private address.

We produced Kaplan-Meier plots for age, sex, deprivation, cancer diagnosis and previous emergency medical admission to examine differences in survival between groups of patients. Age was grouped into age bands (under 60 s and 5-year age bands above this to 85+) for ease of comparison between younger and older persons and detection of non-linear changes with age. Age groups were the same as the previous study for comparability. We modelled survival in days using multivariate Cox proportional hazards models using R3.3.2. Follow-up was 366 days as 2016 was a leap year. We censored patients surviving beyond 366 days from the date of their census emergency admission. We conducted univariate analysis to examine the hazard ratio (HR) associated with the individual variables. Sex, age, deprivation, cancer diagnosis and admission during the previous year were all included in the multivariate Cox regression to determine whether these factors were independent predictors of survival.

There was some evidence of non-proportionality in the Schoenfeld residuals plot for cancer diagnosis (although the HR was always greater than one and quantitative test for a linear trend was non-significant). To test the robustness of our model to this slight non-proportionality and to further investigate differences between patients with and without cancer we repeated the multivariate Cox

**Table 1** Characteristics of patient cohort and mortality rates

| | No. of Admissions | % | Deaths within 7 days | % | Deaths within 30 days | % | Deaths within 3 months | % | Deaths within 6 months | % | Deaths within 9 months | % | Deaths within 1 year | % |
|---|---|---|---|---|---|---|---|---|---|---|---|---|---|---|
| **Age group** | | | | | | | | | | | | | | |
| 0–59 | 3915 | 37.4 | 27 | 0.7 | 80 | 2.0 | 179 | 4.6 | 227 | 5.8 | 277 | 7.1 | 309 | 7.9 |
| 60–64 | 710 | 6.8 | 12 | 1.7 | 35 | 4.9 | 79 | 11.1 | 104 | 14.6 | 125 | 17.6 | 132 | 18.6 |
| 65–69 | 926 | 8.8 | 27 | 2.9 | 64 | 6.9 | 133 | 14.4 | 172 | 18.6 | 207 | 22.4 | 234 | 25.3 |
| 70–74 | 1066 | 10.2 | 39 | 3.7 | 98 | 9.2 | 192 | 18.0 | 229 | 21.5 | 270 | 25.3 | 297 | 27.9 |
| 75–79 | 1175 | 11.2 | 27 | 2.3 | 89 | 7.6 | 203 | 17.3 | 263 | 22.4 | 315 | 26.8 | 351 | 29.9 |
| 80–84 | 1213 | 11.6 | 53 | 4.4 | 125 | 10.3 | 243 | 20.0 | 320 | 26.4 | 376 | 31.0 | 406 | 33.5 |
| 85+ | 1472 | 14.0 | 88 | 6.0 | 202 | 13.7 | 369 | 25.1 | 465 | 31.6 | 560 | 38.0 | 617 | 41.9 |
| **Sex** | | | | | | | | | | | | | | |
| Women | 5463 | 52.1 | 145 | 2.7 | 332 | 6.1 | 666 | 12.3 | 857 | 15.7 | 1031 | 18.9 | 1138 | 20.8 |
| Men | 5014 | 47.9 | 128 | 2.6 | 361 | 7.2 | 732 | 14.6 | 923 | 18.4 | 1099 | 21.9 | 1208 | 24.1 |
| **SIMD** | | | | | | | | | | | | | | |
| SIMD 1 (most) | 3092 | 29.5 | 78 | 2.5 | 204 | 6.6 | 372 | 12.0 | 473 | 15.3 | 564 | 18.2 | 621 | 20.1 |
| SIMD 2 | 2478 | 23.7 | 66 | 2.7 | 155 | 6.3 | 342 | 13.8 | 436 | 17.6 | 519 | 20.9 | 574 | 23.2 |
| SIMD 3 | 1844 | 17.6 | 47 | 2.5 | 119 | 6.5 | 255 | 13.8 | 309 | 16.8 | 374 | 20.3 | 411 | 22.3 |
| SIMD 4 | 1667 | 15.9 | 43 | 2.6 | 121 | 7.3 | 236 | 14.2 | 305 | 18.3 | 370 | 22.2 | 397 | 23.8 |
| SIMD 5 (least) | 1396 | 13.3 | 39 | 2.8 | 94 | 6.7 | 193 | 13.8 | 257 | 18.4 | 303 | 21.7 | 343 | 24.6 |
| **Cancer diagnosis** | | | | | | | | | | | | | | |
| No | 8912 | 85.1 | 187 | 2.1 | 414 | 4.6 | 799 | 9.0 | 1055 | 11.8 | 1318 | 14.8 | 1476 | 16.6 |
| Yes | 1565 | 14.9 | 86 | 5.5 | 279 | 17.8 | 599 | 38.3 | 725 | 46.3 | 812 | 51.9 | 870 | 55.6 |
| **Emergency admission in previous year** | | | | | | | | | | | | | | |
| No | 5410 | 51.6 | 98 | 1.8 | 250 | 4.6 | 507 | 9.4 | 621 | 11.5 | 747 | 13.8 | 819 | 15.1 |
| Yes | 5067 | 48.4 | 175 | 3.5 | 443 | 8.7 | 891 | 17.6 | 1159 | 22.9 | 1383 | 27.3 | 1527 | 30.1 |
| Total | 10477 | 100.0 | 273 | 2.6 | 693 | 6.6 | 1398 | 13.3 | 1780 | 17.0 | 2130 | 20.3 | 2346 | 22.4 |

SIMD, Scottish Index of Multiple Deprivation.

regression analysis. The HR was non-proportional for emergency admissions, in patients without cancer and in all patients combined. The trend fitted a linear function of log time so it was possible to fit an interaction term to account for this. Confidence intervals (CI) for the interaction term were computed using the delta method.

### Patient and public involvement

Patients and public were not involved in the design or preparation of this study

### RESULTS

We identified 10 477 patients with emergency admissions to medicine during the 2-week period, 18–31 March 2015 (after exclusions noted in methods). There were more women (52.1%) than men (47.9%). Most patients were 60 years or older (62.6%), and 14.0% were 85 or older. A greater proportion of admissions came from the most deprived areas (SIMD 1, 29.5%) compared with the least deprived areas (SIMD 5, 13.3%). A total of 1565 (14.9%) patients had been given a cancer diagnosis in the previous 5 years. Just under half (5067 patients, 48.4%) had required one or more emergency admissions in the year before the census admission (table 1).

### Deaths following the census admission

There were 2346 deaths (22.4% mortality) in the year following the census admission. Table 1 shows the number and percentage of deaths that occurred at different time intervals during the year. Six hundred and ten patients died during the census admission (5.8% of all admissions and 26% of all deaths) while 1736 of the deaths that occurred during the year did so after the patient had been discharged (74% of all deaths). Overall, men were

more likely to die than women (24.1% vs 20.8%) and this higher mortality was demonstrated within each age group. (table 1 and figure 1). Mortality rose steeply with age and was five times higher at 1 year for patients aged 85 years and over compared with those who were under 60 years of age (41.9% vs 7.9%). A slightly lower proportion of patients from the most deprived areas (SIMD 1) died during follow-up (20.1%) compared with the less deprived quintiles (ranging from 22.3% to 24.6%). Patients with cancer had higher mortality than patients without a cancer diagnosis (55.6% v 16.6% mortality at 1 year) as did patients who had been admitted as an emergency during the year prior to the census admission compared with those with no previous emergency admissions (30.1% vs 15.1% mortality) (table 1).

### Cause of death and place of death

Three categories of primary cause of death accounted for almost three-quarters of all deaths in the cohort. These were malignant neoplasms (33.8%, with the most common subgroup being cancer of the trachea, bronchus and lung), circulatory diseases (22.5%, mainly ischaemic heart disease) and respiratory disease (17.9%, mainly chronic lower respiratory tract disease). The most common place of death was a National Health Service (NHS) hospital which accounted for 1594 (67.9%) of the 2346 deaths. The remainder of the deaths occurred either at home or other private address (17.8%) or in a care home or other institution (14.3%).

### Comparison with the general populatio

We calculated age-standardised mortality rates using the 2013 European Standard Population to take account of the differences in the age distribution of our emergency

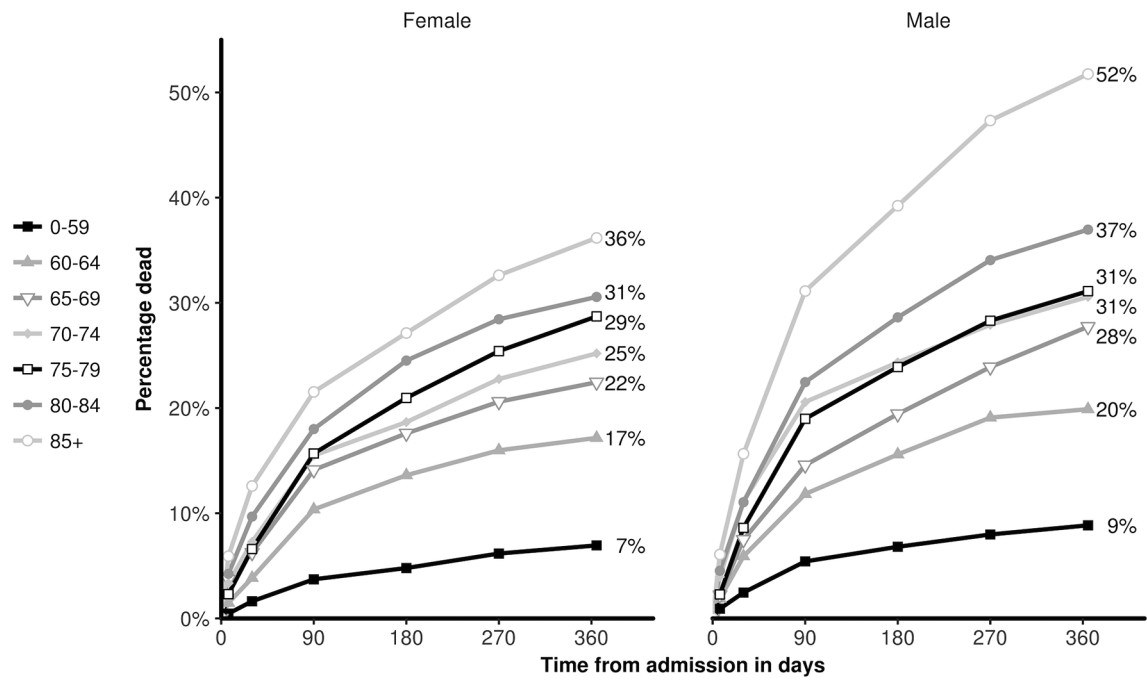

**Figure 1** Cumulative deaths as % of census admissions in male and female patients by age.

medical admissions and the Scottish population. Age-standardised mortality for men was 122.0 per 1000 (95% CI 113.0 to 131.0) which was nine times higher than that of the Scottish population (13.4 per 1000, 95% CI 13.3 to 13.6) from which they were derived. Age-standardised mortality rates for women in our cohort and the general population were 98.9 per 1000 (90.9 to 106.9) and 10.0 (95% CI 9.8 to 10.1) respectively, indicating a 10-fold increase in risk. Results for both sexes combined were 110.4 (95% CI 104.4 to 116.5) for the cohort and 11.7 (95% CI 11.6 to 11.8) for the Scottish population, a ninefold increase in risk.

### Cox regression for all patients in the cohort

Gender, age, deprivation, cancer diagnosis and one or more previous emergency admissions were all risk factors for mortality in the univariate analysis (table 2, figure 2), but in the multivariate analysis, only gender, age, cancer and previous admission were independent risk factors of death (table 2). Men were 1.24 (CI 1.14 to 1.34) times more likely to die than women after adjusting for other risk factors. Older patients had an increased risk of death:

the adjusted HR for those aged over 85 compared with those aged under 60 was 5.74 (CI 4.99 to 6.59). Cancer diagnosis was an important independent predictor of death, with nearly four times increase in risk (HR 3.56, CI 3.27 to 3.88). Emergency admission to hospital in the year prior to the census admission increased risk, although this was non-linear: estimated HR increased from 1.25 (0.93–1.58) on the census day to 1.67 (CI 1.51 to 1.82) at 30 days and 2.05 (CI 1.77 to 2.34) at the end of follow-up. By contrast, deprivation was not a risk factor once other factors were taken into account. A possible explanation for this finding is that patients from the most deprived areas, who were over-represented in our dataset, were less unwell at the time of admission.

### Cox regression for patients with and without cancer separately

Men were more likely to die than women in both cancer and non-cancer patient groups. The effect size was similar with adjusted HR for men compared with women of 1.23 (CI 1.07 to 1.41) and 1.33 (CI 1.20 to 1.47) in patients with and without cancer,

**Table 2** Cox regression analysis for mortality in 1 year of follow-up, for all patients in the cohort

| | No. of deaths | No. of patients | Unadjusted HR (95% CI) | P values | Adjusted HR (95% CI) | P values |
|---|---|---|---|---|---|---|
| **Age** | | | | | | |
| <60 | 309 | 3915 | 1 | – | 1 | – |
| 60–64 | 132 | 710 | 2.51 (2.05 to 3.08) | <0.001 | 2.15 (1.75 to 2.63) | <0.001 |
| 65–69 | 234 | 926 | 3.53 (2.98 to 4.18) | <0.001 | 2.73 (2.30 to 3.24) | <0.001 |
| 70–74 | 297 | 1066 | 4.01 (3.42 to 4.70) | <0.001 | 2.99 (2.54 to 3.51) | <0.001 |
| 75–79 | 351 | 1175 | 4.28 (3.68 to 4.99) | <0.001 | 3.21 (2.75 to 3.74) | <0.001 |
| 80–84 | 406 | 1213 | 4.99 (4.31 to 5.79) | <0.001 | 4.15 (3.57 to 4.82) | <0.001 |
| 85+ | 617 | 1472 | 6.60 (5.76 to 7.57) | <0.001 | 5.74 (4.99 to 6.59) | <0.001 |
| **Sex** | | | | | | |
| Women | 1138 | 5463 | 1 | – | 1 | – |
| Men | 1208 | 5014 | 1.18 (1.09 to 1.28) | <0.001 | 1.24 (1.14 to 1.34) | <0.001 |
| **Deprivation** | | | | | | |
| SIMD 5=least deprived | 343 | 1396 | 1 | – | 1 | – |
| SIMD 4 | 397 | 1667 | 0.97 (0.84 to 1.12) | 0.678 | 1.00 (0.90 to 1.12) | 0.728 |
| SIMD 3 | 411 | 1844 | 0.90 (0.78 to 1.04) | 0.146 | 0.96 (0.85 to 1.09) | 0.676 |
| SIMD 2 | 574 | 2478 | 0.94 (0.82 to 1.07) | 0.342 | 0.96 (0.84 to 1.09) | 0.295 |
| SIMD 1=most deprived | 621 | 3092 | 0.80 (0.70 to 0.91) | <0.001 | 0.93 (0.82 to 1.07) | 0.313 |
| **Cancer diagnosis** | | | | | | |
| No | 1476 | 8912 | 1 | – | – | – |
| Yes | 870 | 1565 | 4.53 (4.16 to 4.92) | <0.001 | 3.56 (3.27 to 3.88) | <0.001 |
| **Emergency admission in previous year** | | | | | | |
| No | 819 | 5410 | 1 | – | 1 | – |
| Yes | 1527 | 5067 | 1.60 (1.24 to 2.08) | <0.001 | 1.25 (0.97 to 1.62) | 0.089 |
| Time*Emergency* | – | – | 1.08 (1.01 to 1.14) | 0.016 | 1.09 (1.02 to 1.15) | 0.006 |

Time function in the interaction was log(t+1) where t is time in days.
SIMD, Scottish Index of Multiple Deprivation.

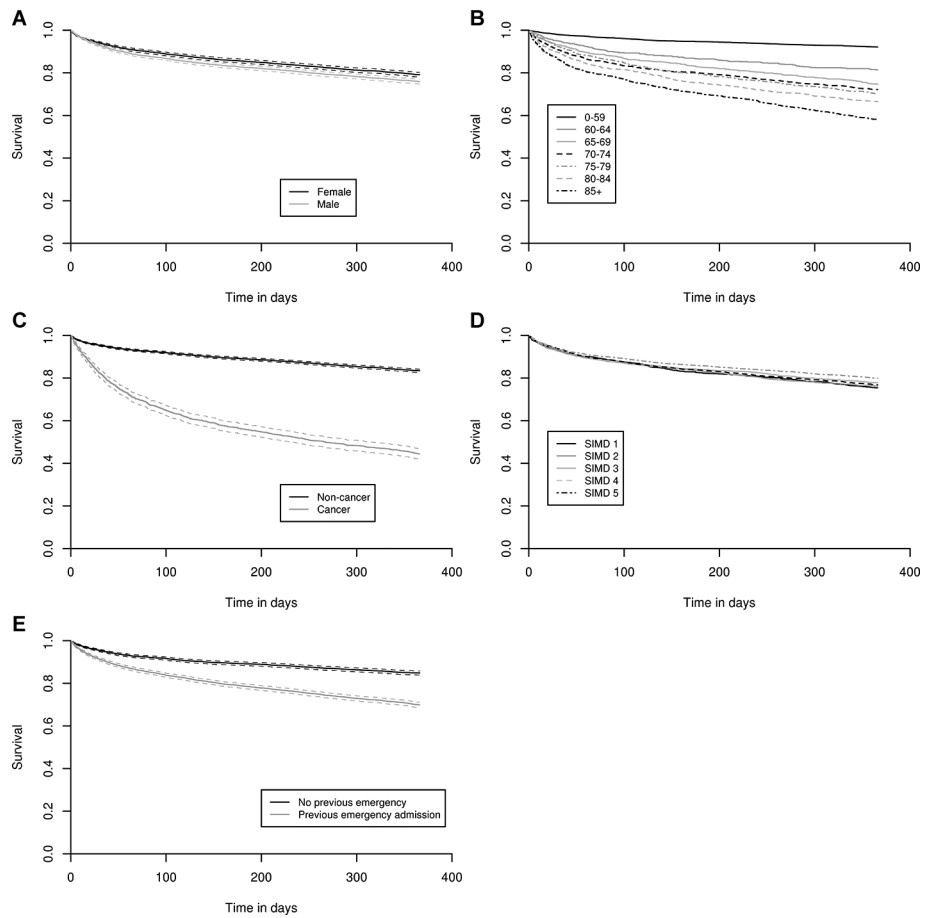

**Figure 2** Kaplan-Meier survival curves of patient groups by (A) sex, (B) age group, (C) cancer diagnosis, (D) Scottish Index of Multiple Deprivation (SIMD) quintile and (E) whether or not patient had one or more emergency admissions in the year prior to the census admission.

respectively. Absolute mortality increased with age in both groups but was always higher in patients with a cancer diagnosis. Compared with those aged under 60 years, patients without cancer over 85 had a higher adjusted HR of 10.16 (CI 8.50 to 12.13) than patients with cancer of the same age (adjusted HR 1.56 (CI 1.23 to 1.98)). This was a consequence of lower absolute mortality in younger patients who did not have cancer (47 per 1000 for patients without cancer under 60 vs 463 per 1000 for patients with cancer under 60). Compared with patients with no emergency admission in the previous year, patients without cancer with one or more previous admissions had an adjusted HR of 1.26 (CI 0.86 to 1.66) on the census day, 1.36 (CI 0.99 to 1.77) at 1 day, 1.86 (CI 1.64 to 2.09) at 30 days and 2.48 (CI 2.07 to 2.48) at 366 days. Barring the hazard on the admission day itself, these were higher ratios than observed for patients with cancer with one or more previous admissions (adjusted HR 1.31, CI 1.14 to 1.51, constant over follow-up time). By contrast, deprivation did not predict outcome in patients either with or without cancer. An online supplementary table showing the Cox regression for patients with and without cancer separately is available on request.

## DISCUSSION

We confirm previous findings in an incident rather than prevalent population of emergency medical admissions to Scottish hospitals in 2015. Over one in five patients died within a year of their census admission with three-quarters of the deaths occurring after rather than during that admission. Mortality rose steeply with age and was five times higher at 1 year for patients aged 85 years and over compared with those who were under 60. Our new findings are that likelihood of death was more closely related to age and to a cancer diagnosis than it was to gender or social deprivation. Over half of all cancer patients died during the 12 months of follow-up. Patients with cancer were more than three times likely to die than patients without a cancer diagnosis. Mortality was also significantly higher among patients who had required one or more emergency admissions in the year before the census admission. Age/sex-standardised mortality for men and women was 9 and 10 times higher, respectively, than the general population from which they were derived.

These findings have important implications for health and social care. Around 550 000 people die in the UK each year. This number is expected to rise to 615 000 deaths per year by 2030.[20] These deaths commonly occur

in hospital and are frequently preceded by one or more emergency hospital admissions. There were over 6 million emergency admissions to NHS hospitals in England[21] and Scotland[22] in 2015–2016. The continuing rise in emergency admissions to hospital[21] likely reflects an increase in life expectancy that is not always healthy life expectancy. The Office for National Statistics has estimated that between 2013 and 2015, UK men at age 65 could expect to live for a further 18.5 years with 10.3 of these years in good health. The corresponding figures for women aged 65 are 20.9 and 11.1 years, respectively. Thus, men and women aged 65 can expect to live just over half of their remaining years in good health.[23] Similar findings have been reported by European[24] and US investigators.[25]

The General Medical Council considers that patients are approaching the end of life when they are likely to die within 12 months. This definition of end of life includes patients whose death is imminent, those with advanced progressive incurable conditions and patients with general frailty and coexisting conditions that mean they are expected to die within 12 months.[7] The Gold Standards Framework Proactive Identification Guidance and the Supportive and Palliative Care Indicators Tool may be used to identify people in the latter two groups whose health is deteriorating and who may be entering the last year of life. Both suggest asking the surprise question ('Would you be surprised if the patient were to die in the next year, months, weeks or days?') and looking for specific clinical indicators of decline relating to the three broad trajectories of illness: cancer, organ failure and frailty.[5 26]

There is now a continuum of interventions possible within the scope of modern medicine with growing interest in the integration of palliative care alongside curative treatments, rehabilitation and the management of long-term conditions. We know that this is often not addressed during emergency medical admission and that doctors do not always feel comfortable making advance or anticipatory care plans with patients and their families despite recommendations that they should do so.[5–9] Up to 26% patients with cancer, 59% of those with organ failure and 34% with frailty do not have advance care plans in place before death.[27] This means that a request to ask 'difficult questions' relating to cardiopulmonary resuscitation and escalation to high dependency or intensive care when a patient is admitted as an emergency to medicine can sometimes cause unintended distress.[28]

If a more modern palliative care orientation is taken, such questions would become subsumed under a broader set of issues relating to the broader goals of the patient. These are eloquently captured by Gawande: What is your understanding of the situation and its potential outcomes? What are your fears and what are your hopes? What are the trade-offs you are willing to make and not willing to make? And what is the course of action that best serves this understanding?[29] All clinicians involved in caring for patients at the end of life have a responsibility to communicate effectively with patients, their families

and members of the multidisciplinary team in order to explore treatment goals and make key decisions.[30 31] Good advance care planning in hospital[32] could mean that the next time a frail older patient becomes unwell, a course of action ensues which does not result in emergency admission to hospital.

We are aware of the limitations to our study. First, if patients emigrated and died abroad during the year of follow-up, then this would underestimate their mortality. We think this is unlikely to be a source of major imprecision. Second, and more importantly, we recognise that our analysis identifies groups of individuals at high risk of death within 1 year and that it cannot and should not be used to predict an individual's risk of dying.

In conclusion, we believe these data may help identify groups of patients admitted to hospital as medical emergencies who are at greatest risk of dying not only during admission but also in the following 12 months. Emergency admission to hospital therefore provides an important opportunity to make advance care plans if appropriate and if such discussions have not already begun. Ultimately, we believe it is wrong to deny the need for an approach at the end of life that might provide care that is more humane and perhaps less costly.[33 34]

**Contributors** CI, EM, LS, RM-A and DC designed the study. EM was responsible for record linkage and statistical analyses. CI wrote the first draft and worked with AR on the second draft. All authors contributed to the final draft.

**Funding** DC is supported by a Wellcome Trust Investigator Award, grant number 103319/Z/13/Z.

**Competing interests** None declared.

**Patient consent** Not required.

**Provenance and peer review** Not commissioned; externally peer reviewed.

**Data sharing statement** Additional data can be accessed in the web appendix. Any queries about source data should be directed to Information Services Division Scotland.

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
