## [Reviewer comments · BMJ Open]

ARTICLE DETAILS

TITLE (PROVISIONAL)	Death within one year among emergency medical admissions to Scottish hospitals: incident cohort study.
AUTHORS	Moore, Emily; Munoz-Arroyo, Rosalia; Schofield, Lauren; Radley, Alice; Clark, David; Isles, Chris

VERSION 1 – REVIEW

REVIEWER	John Kellett Adjunct Associate Professor in Acute and Emergency Medicine at the University of Southern Denmark, Denmark
REVIEW RETURNED	10-Jan-2018

GENERAL COMMENTS	This is a very important paper of great clinical relevance. It needs to be more widely read by my colleagues. I have only one major suggestion: In their introduction the authors state that “for many individuals, admission to hospital is a sentinel event marking the transition to the last year of their lives”. For this to be true they need to provide more information on patients who are admitted for either the first time, or at least the first time in a long time. Although the authors found that on the multivariate analysis only gender, age and cancer were independent predictors of death, I wonder if they examined patients who had not been previously admitted to hospital as a separate categorical group? It appears the approximately half the patients had not been admitted to hospital in the previous year. I would be most interested to know if these patients are a special group. As a junior doctor I was led to believe that a patient (especially an elderly one) who had never been admitted to hospital before was far more likely to die and have a poor prognosis than those who had been frequently sick throughout their lives. Indeed, as the authors have shown, those patients that can tolerate multiple frequent admissions seem less likely to die. From the practicing clinicians point of view I think knowing the 1 year mortality according to age for patients who had never been admitted to hospital before (or within the last year if that is the only data available) would be of value, as this is, I suspect, the category of patients that are most often overlooked with regard to making “end of life” decisions etc. It would also support the claim that “the FIRST admission to hospital is a sentinel event”. Based on the figures available in the author’s paper patients with no admission in the previous year are 0.41 (0.38 – 0.46) less likely die within a year than patients with ANY admission, but still considerably greater than the general public. I find hard to believe that this is not an independent variable. If it isn’t this should be made clear in the text, either as a Result or in the Discussion.
---

REVIEWER	Bernard Silke St James's Hospital and Trinity College, Dublin, Ireland
REVIEW RETURNED	15-Jan-2018

GENERAL COMMENTS	The data in this paper is good and therefore it deserves to be published. However, I have a fundamental problem with the philosophy of this paper. The clinician has to manage uncertainty and beware of unreliable predictions at the individual level. The role of the clinician is not to hasten death, or compromise any reasonable opportunity to improve the outlook for the patient. Most especially not to deny them hope. My standard reply to the patient question ... Is there no hope then, Doctor? There is always hope. Large numbers of hospital inpatients have entered the last year of their lives. This is the starting point of the paper. And of course that is quite true. Of the discharged patients in Ireland following an emergency medical admission, 25% > 60 yr will not survive much beyond 1 year after the acute admission (Kellett data - QJM outcome 1 yr followup). So what is the attitude of the Acute Medicine physician to be - based on these data? The first problem the paper is the actual mortality rate - 3098 (28.8%) patients died during follow-up: 2.9% by 7 days, 8.9% by 30 days. If the 8.9% is a calculated number by episode, it is far too high. The Acute Physicians need to work on that figure, in the interests of the Scottish population, and get it right down. It should be below 5%. Our mortality figures were 12% in 2002 but now are well below 5% and certainly approaching 4.5%. Many other large centres are achieving such figures. But still I accept the authors data, that perhaps 20% of those discharged will be dead at one year following their discharge. But if we accept a 5% mortality figure overall and look at our data between 2002 and 2016. There are about 4300 deaths from > 96000 acute admissions. That is to say, the 96000 patients, of whom 30000 must well be dead by now, died somewhere else ! And we did not admit a plethora of discharged patients from other institutions, who graced us with their final days. So I have a big problem with the application of the population statistics to the care of the emergency medical admission. So now the crunch regarding predictions. One cannot accurately predict which patients are going to be dead at 1 yr from any 30-day data. That is to say if 20% will die by 1 yr, the likelihood, in a Bayesian sense, is low : if an occurrence is not likely before a prediction it is not likely after a prediction, irrespective of the prediction. The PPV will be about 25-30% on average. Most predictions of death based on short-term data will be wrong at an individual level. The majority will still be alive at 1 year. So it is one thing to have an overview, at a population level, of the fate of patients. It is quite another, to attempt to apply that data to the acute patient, with the surrounding uncertainty as to precisely whom (what individuals) will die by 1 year.
--

	The current UK General Medical Council guidance on end-of-life care requires doctors to ensure that death becomes an explicit discussion point when patients are likely to die within 12 months and places a strong emphasis on patient choice rather than 'medical paternalism ... This view is all very fine, if one can accurately predict. But that is simply not possible and the consequence will be doctors incorrectly allocating patient to end of live pathways, when they should be concentrating on measures to improve outcomes How the data of the authors can be translated in a protocol for palliation relevant to the practice of Medicine, I do not know and respectfully point on the problem with attempting to manage uncertainty like this. My view is that one implements palliation for an appropriate population when individual patients can be identified reliably. This may be unsatisfactory to Health Service managers but the dangers of protocols on terminal care being implement too readily are all too evident. So my recommendation to the authors is to report their data as it is. The statistics are good. The great weakness is to attempt to link the data about a high mortality population to palliative care protocols that are not relevant to the majority of hospital admissions who are survivors
--	---

VERSION 1 – AUTHOR RESPONSE

Reviewer: 1

Reviewer Name: John Kellett

Institution and Country: Adjunct Associate Professor in Acute and Emergency Medicine at the University of Southern Denmark, Denmark

Competing Interests: None declared

This is a very important paper of great clinical relevance. It needs to be more widely read by my colleagues.

I have only one major suggestion:

In their introduction the authors state that “for many individuals, admission to hospital is a sentinel event marking the transition to the last year of their lives”. For this to be true they need to provide more information on patients who are admitted for either the first time, or at least the first time in a long time. Although the authors found that on the multivariate analysis only gender, age and cancer were independent predictors of death, I wonder if they examined patients who had not been previously admitted to hospital as a separate categorical group? It appears the approximately half the patients had not been admitted to hospital in the previous year. I would be most interested to know if these patients are a special group. As a junior doctor I was led to believe that a patient (especially an elderly one) who had never been admitted to hospital before was far more likely to die and have a poor prognosis than those who had been frequently sick throughout their lives. Indeed, as the authors have shown, those patients that can tolerate multiple frequent admissions seem less likely to die. From the practicing clinicians point of view I think knowing the 1 year mortality according to age for patients who had never been admitted to hospital before (or within the last year if that is the only data available) would be of value, as this is, I suspect, the category of patients that are most often overlooked with regard to making “end of life” decisions etc. It would also support the claim that “the FIRST admission to hospital is a sentinel event”. Based on the figures available in the author’s paper patients with no admission in the previous year are 0.41 (0.38 – 0.46) less likely die within a year than patients with ANY admission, but still considerably greater than the general public. I find hard to

believe that this is not an independent variable. If it isn't this should be made clear in the text, either as a Result or in the Discussion.

We thank Dr Kellett for this observation. We have rerun the analyses with admitted/not admitted as an emergency in the year preceding the census admission and yes it is an independent variable. Those patients who had required one or more emergency admissions had a significantly worse outcome after adjusting for competing risk factors (we recognise this result is the opposite of what he was expecting).

Reviewer: 2

Reviewer Name: Bernard Silke

Institution and Country: St James's Hospital and Trinity College, Dublin, Ireland

Competing Interests: None Declared

The data in this paper is good and therefore it deserves to be published.

However, I have a fundamental problem with the philosophy of this paper. The clinician has to manage uncertainty and beware of unreliable predictions at the individual level. The role of the clinician is not to hasten death, or compromise any reasonable opportunity to improve the outlook for the patient. Most especially not to deny them hope. My standard reply to the patient question ... Is there no hope then, Doctor?

There is always hope.

We agree the role of the clinician is not to hasten death, or compromise any reasonable opportunity to improve the outlook for the patient. Essentially all we are asking is that doctors make a plan when frail older adults are admitted as emergency to hospital. See new concluding paragraph.

Large numbers of hospital inpatients have entered the last year of their lives. This is the starting point of the paper. And of course that is quite true. Of the discharged patients in Ireland following an emergency medical admission, 25% > 60 yr will not survive much beyond 1 year after the acute admission (Kellett data - QJM outcome 1 yr followup).

The Kellett paper (QJM 2012; 105:847-53) was a study of a subgroup of emergency medical admissions who had a Simple Clinical Score and ECG dispersion mapping within 10-20 minutes of hospital admission. 16.3% died within a year of follow up. Dr Kellett concluded that 'this information should prompt clinicians to initiate discussions on advance care planning and goals of care in their patients'. We agree with this conclusion and now reference this paper in our introduction.

So what is the attitude of the Acute Medicine physician to be - based on these data?

The first problem the paper is the actual mortality rate - 3098 (28.8%) patients died during follow-up: 2.9% by 7 days, 8.9% by 30 days. If the 8.9% is a calculated number by episode, it is far too high. The Acute Physicians need to work on that figure, in the interests of the Scottish population, and get it right down. It should be below 5%. Our mortality figures were 12% in 2002 but now are well below 5% and certainly approaching 4.5%. Many other large centres are achieving such figures. But still I accept the authors data, that perhaps 20% of those discharged will be dead at one year following their discharge.

Silke et al (QJM 2010; 103: 23-32) quote an in hospital 5 day mortality of 6.1%. Our deaths within 7 days ranged from 0.7% for those under 60 to 6.0% for those over 85 which sounds as if it is exactly where it should be. An important finding of ours is that 74% of all deaths occurring during 12 months of follow up occurred after discharge from hospital.

But if we accept a 5% mortality figure overall and look at our data between 2002 and 2016. There are about 4300 deaths from > 96000 acute admissions. That is to say, the 96000 patients, of whom 30000 must well be dead by now, died somewhere else! And we did not admit a plethora of discharged patients from other institutions, who graced us with their final days. So I have a big problem with the application of the population statistics to the care of the emergency medical admission.

We believe Dr Silke is referring to Cournane et al Computational and Mathematical Models in Medicine 2017; doi.org/10.1155/2017/5267864 though we are not sure what point he is making here. We don't think we are attempting to apply population statistics to the care of the emergency medical admission. Happy to discuss further if we have missed the point.

So now the crunch regarding predictions. One cannot accurately predict which patients are going to be dead at 1 yr from any 30-day data. That is to say if 20% will die by 1 yr, the likelihood, in a Bayesian sense, is low : if an occurrence is not likely before a prediction it is not likely after a prediction, irrespective of the prediction. The PPV will be about 25-30% on average. Most predictions of death based on short-term data will be wrong at an individual level. The majority will still be alive at 1 year. So it is one thing to have an overview, at a population level, of the fate of patients. It is quite another, to attempt to apply that data to the acute patient, with the surrounding uncertainty as to precisely whom (what individuals) will die by 1 year.

Agree. We already make the point (in our limitations paragraph) that our analysis identifies groups of individuals at high risk of death within one year and cannot be used to predict an individual's risk of dying. We are happy to emphasise this point by stating "Second, and more importantly, we recognise that our analysis identifies groups of individuals at high risk of death within one year and that it cannot and should not be used to predict an individual's risk of dying".

The current UK General Medical Council guidance on end-of-life care requires doctors to ensure that death becomes an explicit discussion point when patients are likely to die within 12 months and places a strong emphasis on patient choice rather than 'medical paternalism'. This view is all very fine, if one can accurately predict. But that is simply not possible and the consequence will be doctors incorrectly allocating patient to end of live pathways, when they should be concentrating on measures to improve outcomes

Never our intention that all frail older adults should be put on end of life pathways though we do think that doctors should make advance care plans with the patient (if they have capacity) or with the family (if they do not).

How the data of the authors can be translated in a protocol for palliation relevant to the practice of Medicine, I do not know and respectfully point on the problem with attempting to manage uncertainty like this. My view is that one implements palliation for an appropriate population when individual patients can be identified reliably. This may be unsatisfactory to Health Service managers but the dangers of protocols on terminal care being implement too readily are all too evident.

Agree. We have no wish to use our data to create protocols for palliation. All we ask is that doctors consider advance care plans if appropriate and if these have not been discussed before.

So my recommendation to the authors is to report their data as it is. The statistics are good. The great weakness is to attempt to link the data about a high mortality population to palliative care protocols that are not relevant to the majority of hospital admissions who are survivors.

Agree. Addressed as above. Our conclusions are almost identical to those of Dr Kellett in his QJM 2012 paper.

VERSION 2 – REVIEW

REVIEWER	Dr Bernard Silke St James's Hospital and Trinity College, Dublin
REVIEW RETURNED	08-Mar-2018

GENERAL COMMENTS	Revision has addressed issues raised in the first draft. MS is much improved and recommend publication.
---

REVIEWER	John Kellett Nenagh Hospital Ireland
REVIEW RETURNED	16-Mar-2018

GENERAL COMMENTS	This is a most important and valuable paper
---